# Some Properties of Briquettes and Pellets Obtained from the Biomass of Energetic Willow (*Salix viminalis* L.) in Comparison with Those from Oak (*Quercus robur*)

Veronica Dragusanu (Japalela) [1], Aurel Lunguleasa [1,*], Cosmin Spirchez [1] and Cezar Scriba [2]

[1] Wood Processing and Design Wooden Product Department, Transilvania University of Brasov, 29 Street Eroilor, 500036 Brasov, Romania; tamara.dragusanu@unitbv.ro (V.D.); cosmin.spirchez@unitbv.ro (C.S.)

[2] Forestry Operations, Forest Management and Land Surveying Department, Transilvania University of Brasov, 29 Street Eroilor, 500036 Brasov, Romania; caesarus@unitbv.ro

* Correspondence: lunga@unitbv.ro

**Abstract:** Fast-growing species have been increasingly developed in recent years, and among them, those cultivated to obtain combustible woody biomass have shown rapid development. The purpose of this research study is to highlight the properties of the briquettes and pellets obtained from energetic willow compared to the briquettes and pellets obtained from oak biomass. Methodologies have been based on international standards and were used to find the physical, mechanical, and calorific properties of the two types of briquettes and pellets. The results did not highlight a significant difference between the two categories of briquettes and pellets obtained from the two hardwood species (energetic willow and oak). Characteristics such as the calorific value were 20.7 MJ/kg for native pellets and 21.43 MJ/kg for torrefied pellets of energetic willow, as well as the compressive strength of 1.02 N/mm$^2$, surpassed the same characteristics of briquettes and pellets obtained from oak biomass. Other characteristics of energetic willows, such as energetic density of $18.0 \times 10^3$ MJ/m$^3$, splitting strength of 0.08 N/mm$^2$, shear strength of 0.86 N/mm$^2$, and abrasion of 1.92%, were favorably related to the oak biomass. The ecological analysis highlighted the high potential of the ecological willow in a period when the quantities of carbon dioxide released into the atmosphere by human activities are very high, and its sequestration by existing forests is insufficient. As a general conclusion of this research study, it can be stated that the two categories of briquettes and pellets obtained from the woody biomass of the energetic willow and oak species have similar characteristics, which can be used separately or together in ecological and sustainable combustion.

**Keywords:** briquette; pellet; CIELab space; density; calorific value; energetic density; ash content

## 1. Introduction

The main use of lignocellulosic biomass, regardless of its nature and origin, is in the field of energy production. The sources for obtaining it are varied, such as the woody remains from the exploitation of forests and wood processing factories, but also from forestry and agricultural crops dedicated exclusively to obtaining energetic biomass. Among these specially dedicated crops for biomass are listed woody plants with a short rotation cycle (willow, poplar, acacia, etc.), with a high potential to obtain a secure income in the short term [1] but also to take over and sequester of carbon from the atmosphere within a short period of time. A plantation with ordinary woody species of trees needs, on average, 90–160 years to reach maturity, but a plantation with fast-growing species needs only 25–35 years. Therefore, in addition to the fact that plantations with fast-growing species recover their investment faster, they could obtain a (72–78%) better economic efficiency [1]. In addition to the fact that the short-rotation crops are a renewable source of ecological energy [2,3], some fast-growing species also have a role in the remediation of soil areas degraded by heavy metals [4–6] and/or with high moisture content in the soil, improving

environmental pollution problems, recycling of wastewater [7], and sustaining the development of rural localities. This category of energetic crops could include energetic willow (*Salix viminalis*), elephant grass (*Mischantus giganteum*), Chinese reed (*Miscanthus sinensis*), Pampa's grass (*Cortaderia Rosea*), sorghum (*Sorghum halepense*), Sudan grass (*Sorghum sudanense*), etc. [6].

Worldwide, there are over 300 species of willow. The energetic willow (*Salix viminalis*) is an agricultural crop plant that is woody and shrubby, located in agricultural areas and less often in forest habitats. In addition to the role of restoring the productive circuit of some highly degraded lands such as tailings dumps, former sites of chemical plants, and heavily eroded, saline or sandy soils, these woody species produce considerable biomass (about 35 t/year/ha wet biomass, and when some specific irrigation and fertilization treatments are applied, the biomass increase up to 60 t/ha/year) which is available in each growing year [6]. In addition, this woody species has a fast growth cycle with height gains of 3–3.5 cm/day. Its calorific value is 18,224 kJ/kg for 3-year-old sticks (with an average height of 7 m and a diameter at the base of 8 cm) and 18,265 MJ/kg for the bark of the same sticks [6]. The energetic willow contains a large amount of salicylic acid, which helps in a rapid loss of moisture up to 14%–16%, with advantages in its processing without artificial drying and open storage for a long period of time without biological degradation. The harvesting of thin stems is conducted with current agricultural machinery only during the period of vegetative rest after November, such as combine harvesters, tractors, and trailers, when the park of agricultural machinery is under conservation, and the willow leaves have fallen to the ground and can create a fertilizing layer for the soil [8,9]. This woody biomass, in the form of wood chips, chops, briquettes, and pellets, represents a cheaper alternative to ordinary firewood. The energy obtained from energetic willow is a renewable one, the vegetation period of a plantation being 25–30 years, starting with the third year after planting [10–12]. A calorific value of 19–21 MJ/kg, equal to 5.5–6.1 kWh/kg, is ideal for obtaining briquettes and pellets [6,7]. Many authors have analyzed impacts on climate change [13], financial analysis [14], energetic impact [15], biofuel potential [16], yield [17–20], establishing of surface cover [21,22], biomass production [23], trait and genome [24,25], potential [26], wood quality [27], briquetting [28], genetic structure and diversity [29,30], structure [31], and prognosis [32], all of these being obtained on energetic willow plantation. The energetic willow is planted on the ground in such a way that two adjacent rows have a distance of 750 mm between them, and a greater distance of 900 mm between four rows is necessary for moving the harvesting installations [6,21].

Energetic crops represent an alternative to fossil fuels; it is estimated that in the EU-27 will be a potential of 47 Mtoe/year before 2025, and after 2025 this value will exceed 138 Mtoe/year, i.e., namely 14% of the total energy consumption at European level. It is also stated that an area of 13.2 Mha is available for energetic crops before 2025, and after 2025 this will exceed 26.2 Mha [33].

The biomass obtained when harvesting the energetic willow could be stored and used in the form of briquettes and pellets [34]. Going further, other authors [35,36] have improved the energetic and hydrophobic performances of biomass through a torrefaction thermal process. Based on the results obtained at temperatures above 200 °C, the torrefaction process of the woody biomass contributed to an increase in the calorific value by up to 60% and a decrease in the hydrophilicity by up to 20%. It was also specified that at the European level, renewable energy represents 24% of total energy consumption, and the consumption of wood and wood waste represents over 68% of biomass. A small part, namely less than 1% of this woody biomass, is derived from energetic willow crops [33].

Lignocellulosic briquettes and pellets are solid fuel products whose main advantage is the densification of small lignocellulosic biomass up to a density of 800–1250 kg/m$^3$ [37–39]. They maintain the advantages of lignocellulosic biomass, including those related to renewability [40], environmental friendliness [41], and neutrality of carbon dioxide emission [42]. In addition, briquettes obtained from energetic crops (miscanthus, energy willow, sorghum, etc.) [43] bring an important contribution to the natural environment by elimi-

nating oxygen in the atmosphere [44] and sequestering carbon dioxide in each vegetative year [6,45]. Other economic and ecological effects of biomass briquetting are presented by other researchers [46,47].

Objectives: If previous studies in the field refer to particular aspects of the energetic willow, the main objective of the research is to correlate its ecological, physical, and calorific properties. Particularly, the purpose of this research is to analyze the properties of the briquettes and pellets obtained from the biomass of the energetic willow in order to use it effectively. The physical-mechanical properties, such as the density, breaking resistance of the briquettes and shearing resistance of the pellets, and calorific properties, such as calorific value, calorific density, and ash content, will be analyzed. Some ecological aspects of energetic willow will also be emphasized. In addition, to observe the position of the energetic willow biomass within the whole lignocellulosic biomass, a comparison was made between the briquettes and pellets obtained from the energetic willow biomass and that of the oak waste, both in the native and torrefied state.

## 2. Materials and Methods

### 2.1. Ecological Aspects

Similar to any tree or forest, the cultivation of the energetic willow brings multiple additional ecological benefits, two of which are more important, namely the sequestration of carbon dioxide from the atmosphere and the elimination of oxygen through the photosynthesis process [13,48,49]. The method for determining the amount of carbon dioxide sequestered in energetic willow crops is similar to that for trees in forests (as oak species could be) and contains several steps [13], as follows:

— Determining the weight of the green biomass cutting from the part above the ground, using the following relationship for the group of twigs:

$$M1 = \frac{\pi \cdot d^2}{4} \times H \times De \times n \ [\text{kg}] \tag{1}$$

where: $D$ is the average diameter of a stem of the shoot, in m; $H$ is the average height of the shoot stem, in m; $De$ is the density of green wood, in kg/m$^3$; $n$ is the number of sticks resulting from a shoot.

— The addition of the woody part corresponding to the root, which corresponds to about 10% of the area above the ground, respectively, the total value of the woody part will be:

$$M2 = 1.1 \times M1 \ [\text{kg}] \tag{2}$$

— Determination of the absolute dry mass of the woody mass resulting from a stump group, taking into account that 72.5% is dry mass and 27.5% is water in different forms (liquid, vapor, and chemically dissociated), that is, this will be:

$$M3 = 1.2 \times M1 \times 0.725 \ [\text{kg}] \tag{3}$$

— Determination of the carbon mass in the wood of the energetic willow, considering that the carbon content for the energetic willow is 48.4% [50,51], which means a mass of:

$$M4 = 1.2 \times M1 \times 0.725 \times 0.484 \ [\text{kg}] \tag{4}$$

— Determination of the sequestered carbon dioxide content in the woody part of the biomass. It is taken into account that $CO_2$ has one carbon molecule and two other oxygen molecules, the atomic mass of carbon is 12, and that of oxygen is 16. Consequently, the mass of $CO_2$ will be a ratio between the atomic mass of all $CO_2$ and the atomic mass of carbon, respectively, 44/12 = 3.67. Therefore, to determine the mass of carbon dioxide sequestered in trees, the previous mass of carbon M4 will be multiplied by 3.67, which will be:

$$M5 = 3.67 \times M4 \ [\text{kg}] \tag{5}$$

— For an energetic willow crop, the amount of sequestered carbon dioxide per surface unit is calculated with the following relationship:

$$M6 = n1 \times M5 \ [kg \ CO_2/ha] \tag{6}$$

where: n1 represents the number of cuttings existing on one hectare of energetic willow culture.

So, taking into account all variables, the general relationship will be:

$$Mg = \pi \times D^2 \times H \times De \times n \times 0.725 \times 0.484 \times n1 \times 3.67 \times 1.2 \times 0.25 \ [kg \ CO_2/ha] \tag{7}$$

*2.2. Granulometry of the Energetic Willow and Oak Crushed Material*

The willow biomass was taken from an energetic willow plantation, and the oak biomass was taken from a circular saw in the form of sawdust. The granulometry of the small material characterizes its dimensions, depending on which the obtained pellets and briquettes could have better or worse physical–mechanical characteristics. Both types of small materials were sorted with a $5 \times 5$ mm sieve in order to have appropriate and homogeneous sizes. The main purpose of this determination was to determine the different fractions of the small material because an increased percentage of fractions with large sizes will determine a low density and a high breaking strength of the briquettes and pellets, and an increased percentage of the fraction with small sizes will lead to obtaining products with high densities and reduced resistance. For this test, an electrical vibrating device was used with sieve sizes of $4 \times 4$, $3.13 \times 3.13$, $2 \times 2$, $1.25 \times 1.25$, $0.8 \times 0.8$ and $0.4 \times 0.4$. The granulometry of the material was determined by the sieving method. What passed through the last lower sieve was called "Rest." A total of 6 randomly extracted samples of 30 g were used from each type of biomass (oak and energetic willow), and 12 samples were made. In order to obtain the values of the masses (determined with EWJ 600-2M Kern, Merck KGaA, Darmstadt, Germany balance) and the participation percentages for each sieve, the arithmetic mean was performed for the values obtained for each of the 6 samples. For example, the participation percentage of the resulting fraction above the $2 \times 2$ mm sieve was calculated with the following calculation relationship:

$$P_{2 \times 2} = \frac{\frac{1}{6} \sum_{i=1}^{6} m_{2 \times 2}}{m_p} \times 100 \ [\%] \tag{8}$$

where: $m_{2 \times 2}$ is the mass of the fraction that remained above the mesh sieve $2 \times 2$ mm; $m_p$—the mass of the sample taken into consideration for the test, in g.

*2.3. Obtaining Briquettes and Pellets*

The shredded material had an absolute moisture content of $12 \pm 1\%$, determined by the gravimetric weighing-drying-weighing principle [52,53]. Briquetting was carried out on a Gold Star type hydraulic piston machine (Brasov, Romania) with a capacity of 500 kg/h, and pelletizing was carried out on a Sarras mechanical press (Sarras group, Brasov, Romania) with a capacity of 40 kg/h. For a better choice of briquettes and pellets, the briquettes and pellets obtained in the first 5 min of operation were removed from this study. In addition, for better identification of each piece during testing, each sample had an identification number and was placed on flat white support in an order that was maintained throughout the all-testing period.

*2.4. Unit Density of Briquettes and Pellets*

Before this determination, all briquettes and pellets were conditioned at a temperature of 20 °C and 55% air humidity for 24 h in order to stabilize the moisture content at $10 \pm 1\%$. The individual density of briquettes and pellets was determined as the ratio between their mass and volume (DIN 51731: 1996) [54]. Taking into account the cylindrical shape of the briquettes and pellets, the density determination relationship was the following (Equation (9)):

$$\rho = \frac{4 \cdot m}{\pi \cdot d^2 \cdot l} \times 10^6 \ [kg/m^3] \tag{1}$$

where: $m$ is the mass of the sample, in g; $d$ is the diameter of the sample, in mm; $l$ is the length of the sample, in mm.

The bulk density of the pellets was determined according to EN 15103:2009 [55] by using a cylindrical vessel with a known internal volume and weighing the vessel filled with pellets. Eight replicates of this test were used to obtain a consistent mean and standard deviation of statistical results.

### 2.5. The Pellet Torrefaction Treatment

The torrefaction treatment was applied to energetic willow and oak pellets, using temperatures of 180, 200, and 220 °C and 3 treatment times of 1, 2, and 3 h [35]. The main purpose of this determination was to improve the calorific characteristics of the pellets. During torrefaction, part of the wood hemicelluloses is damaged, thereby increasing the calorific value but especially the energetic density through the loss of mass. Additionally, the torrefied pellets become more stable (moisture absorption is reduced) and are sterilized (degradation is more difficult). The higher temperature and duration, the more advantages of torrefaction [41,42]. A Memmert-type laboratory oven (Carbolite Gero Ltd., Hope Valley, UK) was used with the air inlet valve closed in order to eliminate the possibility of oxidation and self-ignition of the samples. Prior to treatment, all the samples were dried for 4 h in the same laboratory oven at a temperature of 105 °C. After drying, the samples were weighed with a precision electronic balance type EWJ 600-2M Kern (Merck KGaA, Darmstadt, Germany), recording their initial and final mass at the end of the treatment period. Based on the two weights, it was possible to determine the mass loss during torrefaction with the help of the following relationship (Equation (10)):

$$ML = \frac{M_i - M_f}{M_i} \times 100 \; [\%] \tag{10}$$

where: $ML$ is the mass loss in %; $M_i$ is the initial mass of samples in g; $M_f$ is the final mass of samples in g.

Eight valid samples for each type of pellet were used to determine the statistical parameters of this test. The biomass briquettes were not torrefied due to their low compressibility and compatibility, which led to their disintegration during the treatment.

### 2.6. Color Determination of Native and Torrefied Pellets with CIELab Colorimetric Space

This colorimetric space is quantified by 3 distinct parameters, L*, a*, and b*. The L* axis represents lightness and has the value zero for black and one hundred for white; between these values, there were a series of shades of gray. The a* axis refers to the green-red opposition, with negative values towards green and positive values towards red. The b* axis quantifies the blue–yellow opposition, with negative values towards the blue zone and positive values towards the yellow zone [56].

### 2.7. Calorific Value and Energetic Density of Briquettes and Pellets

The calorific value of briquettes and pellets of *Salix viminalis* and oak was determined on dry and compact samples with a weight of 0.8 ± 0.1 g, taken from briquettes or pellets [57]. An XRY-1C oxygen bomb calorimeter (Shanghai Geological Ltd., Shanghai, China), provided with its own test software, was used. Before testing, the calorimeter was calibrated by using a pill of 1 g of benzoic acid with a known calorific value of 26,454 kJ/kg, thus finding the calorimetric coefficient k from the relationship for determining the high calorific value (*HCV*) (Equation (11)):

$$HCV = \frac{k \cdot (T_f - T_i) - Q_{ct} - Q_{nw} - Q_a}{m} \; [\frac{\text{kJ}}{\text{kg}}] \tag{11}$$

where: $T_f$ is the final temperature in °C; $T_i$ is the initial temperature in °C; $Q_{ct}$ is the amount of heat released during combustion by the cotton thread in kJ; $Q_{nw}$ is the amount of heat

of the nickel wire in kJ; $Qa$ is the amount of heat given by the nitric acid produced during combustion, in kJ.

Eight valid tests were carried out for each type of material [6,28,33] in order to obtain acceptable and significant values. Each test provided values with a high calorific value, low calorific value, burning time, and evolution of the temperature in the 3 stages of the test (fore, main, and after).

Energetic density expresses the calorific energy obtained from each cubic meter of the briquettes or pellets. It was obtained by multiplying the calorific value by briquette/pellet density with the next relationships (Equation (12)):

$$ED = HCV \times \rho_{b/p} \ \left[ {}^{MJ}\!/_{m^3} \right] \tag{12}$$

where: $ED$ is energetic density in MJ/m$^3$; $HCV$ is the high calorific value in MJ/kg; $\rho_{b/p}$ is the unit density of briquette or pellet in kg/m$^3$.

### 2.8. Ash Content

In order to determine the ash content, 3–5 g of crushed material was taken, which was obtained during the sorting procedure, with the help of the $1 \times 1$ mm sieve. This material was dried in an oven at 105 °C for 30 min up to a constant mass in order to obtain an absolutely dry mass. Next, this dry material was deposited on a crucible and weighed with an analytical balance type EWJ 600-2M Kern (Merck KGaA, Darmstadt, Germany) with a precision of 2 decimals. The crucibles with the dried material were placed in a calcination furnace at a temperature of 750 °C (ASTM E1755-01: 2020; ISO 2171: 2007) [58,59] for about 40 min. Calcination was considered complete when the ash became light gray, without traces of sparks or non-calcified material. At that moment, the crucible with calcined ash was extracted from the furnace, cooled in a desiccator, and weighed with the same high-precision balance. The ash content was determined with the following calculation relationship:

$$A_c = \frac{mc_f - mc}{mc_i - mc} \times 100 \ [\%] \tag{13}$$

where: $m_{ci}$ is the initial mass of the crucible with the sample of dry crushed material in g; $m_{cf}$ is the final mass of the crucible with calcined ash in g; $m_c$ with the mass of the empty crucible in g.

At least 6 valid samples were taken into consideration for each type of sample, obtained from energetic willow and oak biomass.

### 2.9. The Compressive Strength of Briquettes

Before the determination, all briquettes were conditioned at a temperature of 20 °C and a relative air humidity of 65% until obtaining a moisture content of $10 \pm 1\%$. Then, their average length and average diameter were determined. For this determination, the briquettes made of energetic willow and oak were inserted between the two plateaus of the universal WA testing machine (TE Force speed Corporation, Jinan, China) and subjected to the action of compression until they broke. The maximum breaking force was recorded for each briquette, after which the compression breaking resistance was determined as a ratio between the force and the breaking area (Equation (14)):

$$\sigma_c = \frac{F_{max}}{d \cdot l} \ \left[ \frac{N}{mm^2} \right] \tag{14}$$

where: $F_{max}$ is the maximum force of briquette breakage in N; $d$ is the diameter of briquettes in mm; $l$ is the length of briquette samples in mm.

Fifteen valid tests were taken into consideration in order to obtain statistical parameters with a confidence interval of 95%.

### 2.10. Splitting Resistance of Briquettes (Perpendicular and Parallel to the Length)

Even if the briquettes are reconstituted engineering products, similar to the splitting resistance of wood along the longitudinal plane of minimum strength, the splitting resistance of the briquettes was taken into consideration. This resistance was made perpendicular and parallel to the length of the briquettes. To perform the splitting test, the WA-type universal testing machine (TE Force Speed Corporation, Jinan, China) was used, with a specially designed splitting device. The device consisted of a knife with a tip having an angle of 76 degrees and a radius of roundness of 1 mm (in order not to cut the briquette but only to split it). The relations for determining the splitting resistance were the following (Equation (15)):

$$\sigma_{spar} = \frac{F_{max1}}{d \cdot l} \left[\frac{N}{mm^2}\right] \sigma_{sper} = \frac{4 \cdot F_{max2}}{\pi \cdot d^2} \left[\frac{N}{mm^2}\right] \tag{15}$$

where: $\sigma_{spar}$ is the resistance to splitting parallel to the length of the briquettes in N/mm²; $\sigma_{sper}$ is the resistance to splitting perpendicular to the length of the specimen in N/mm²; $F_{max1}$ is the maximum force of the parallel resistance in N; $d$ is the diameter of briquettes in mm; $l$ is the length of briquettes in mm; $F_{max2}$ is the maximum force of splitting strength perpendicular to the length of briquettes in N.

A total of 10 briquette specimens were tested for this determination, both for the longitudinal and transverse splitting of briquettes.

### 2.11. The Pellet Shearing Strength

Shearing of pellets is a current problem encountered during storage, transport, and use. In this way, during the combustion process, they become shorter, with serious repercussions on the management of the combustion process. For this test, a universal WA-type testing machine and two shearing devices (metal plates) fixed on the two arms of the compression machine were used. The upper device had the role of shearing, having an angle at the top of about 80 degrees so as not to cut the pellets, and the lower one was provided with five holes with a diameter of $8^{+0.2}$ mm to support the pellets during shearing. The test was considered finished when the shear force dropped suddenly. The shear strength relationship was determined by the next relationship (Equation (16)).

$$\tau_s = \frac{4 \cdot F_{max}}{5 \cdot \pi \cdot d^2} \left[\frac{N}{mm^2}\right] \tag{16}$$

where: $F_{max}$ is the maximum force of breaking in N/mm²; $d$ is the diameter of the pellet in mm.

### 2.12. The Briquettes Abrasion

The briquettes were abraded by placing them on the electrical vibrating device (Xinxiang Gaofu Machinery Co., Ltd., Xinxiang, China) with a 2 × 2 mm sieve. The briquettes that were used had a constant moisture content of 10 ± 1% and a maximum height of 30 mm in order to obtain sufficient space for the movement of the briquettes on the wear and sorting sieve. The briquette abrasion was determined by the next relationship (Equation (17)):

$$Ab = \frac{m_{us}}{m_s} \times 100 \; [\%] \tag{17}$$

where: $Ab$ is the abrasion of briquettes in %; $m_{us}$ is the mass of sawdust sifted and collected under the 2 × 2 mm sieve in g; $m_s$ is the mass of initial samples in g.

A total of 8 laboratory tests were performed.

### 2.13. Statistical Analysis

An arithmetic mean and a standard deviation were calculated for each group of tested values. In addition, the standard deviations, the regression equations, and the coefficient of determination R² were identified on the Microsoft Excel 2019 (Microsoft Corp., Redmond, WA, USA) graphs. With the help of the statistical analysis program Minitab 18 (Penn State

University, State College, PA, USA) and its specific graphs, the upper and lower limits of the calculated values were identified for a confidence interval of 95% or an alpha error of 0.05.

## 3. Results

### 3.1. Ecological Aspects of the Willow/Oak Plantation

Following the calculations made according to the methodology presented in Section 2.1 above, a mature cutting aged 10 years could absorb about 20 kg of carbon dioxide per year from the atmosphere. Taking into account that saplings under the age of 3 years have reduced foliage and will absorb less $CO_2$, it results that in an average life of 100 years, four saplings could absorb about one ton of $CO_2$. Reported at the surface of a plantation, the sequestering of $CO_2$ was about 4000 tons of $CO_2$. This value is compared with the amount of carbon dioxide eliminated by human activity of, about 40 billion tons of $CO_2$ every year [6]. Extrapolating to the level of the planet's three thousand billion trees, it is found that each human life would need about four hundred twenty trees. Relating it to the world's forests and the number of world's trees, a negative ratio is obtained, i.e., for an emission of 40 billion tons of carbon dioxide, the forests will absorb only 30 billion tons of carbon dioxide, with 10 billion tons of carbon dioxide remaining in the atmosphere and leading to global warming through the greenhouse effect. Therefore, another thousand billion trees would be needed, or taking into account the existence of 5000 trees per hectare, a forested area of about 500 million hectares would be needed. The essential problem of the planet nowadays is that every year about 5% of the world's forests are lost [40–43] by returning deforested lands to agriculture, and the amount of carbon dioxide increases yearly. The energetic willow is considered an agricultural crop, which is why it can compensate for the shortage of trees. In addition to the $CO_2$ emissions that was described above, it could be added a considerable amount of methane gas, eliminated in particular by agricultural farms all over the world [46,47], but also by the degradation of fallen trees due to storms and other natural weather/fire problems [47]). Methane gas released into the atmosphere has a much greater influence than $CO_2$ on global warming.

Regarding the release of oxygen during the photosynthesis process, the analysis is similar to that of any fast-growing woody species. The crown of a mature cutting produces, on average, 117.9 kg of oxygen each year. It is known that a person needs 9.5 tons of air per year, or considering the composition of air with 23% oxygen and human breathing uses only 25% of this oxygen, it means that a person needs 3.85 t oxygen per year. From these considerations, it follows that a person needs about 30 trees to release the oxygen necessary for breathing. Hence, the slogan that every man must plant a tree in his life must be multiplied 30 times nowadays. It is observed that from this point of view, in total—without taking into account the diversity of the fauna and the different climates from one area of the globe to another—the world's forests still provide the oxygen needed for human life.

### 3.2. Bulk Density of the Sawdust of the Two Types of Biomasses

Bulk density was determined as a ratio between the mass and the volume of the crushed material disposed into the vessel. The volume was determined using a cylindrical vessel with an inner diameter of 20.17 mm and a height of 42.67 mm, respectively, with a volume of 1342.11 cm$^3$. The mass of the material contained in the cylindrical vessel was determined as the difference between the mass of the cylinder with material and the mass of the empty vessel, with the help of an analytical balance with a precision of two decimals. For a good placement of the material in the analysis vessel, it was vibrated for 3 min. Based on the 10 tests performed both on the small material of energetic willow and oak, bulk densities of 480.2 kg/m$^3$ for oak and 209.9 kg/m$^3$ for energetic willow were obtained. Taking into account the effective density of the two wood species of 675 kg/m$^3$ for oak and 400 kg/m$^3$ for willow [47], compacting coefficients of 1.41 for oak and 1.97 for energetic willow were obtained. Therefore, the shredded material of willow was considered much

looser than that of oak due to the low density of the species, which had repercussions on the increase in compaction in briquettes and pellets.

### 3.3. Granulometry of Wood Particles

Due to the use of the same 5 × 5 mm sorting sieve of sawdust before briquetting and pelletizing, the two granulometry curves were almost similar, as can be seen in Figure 1. A slight shift to the right in the granulometric curve of the energetic willow, or the upward shift in the oak curve, was due to the greater weight of the oak chips.

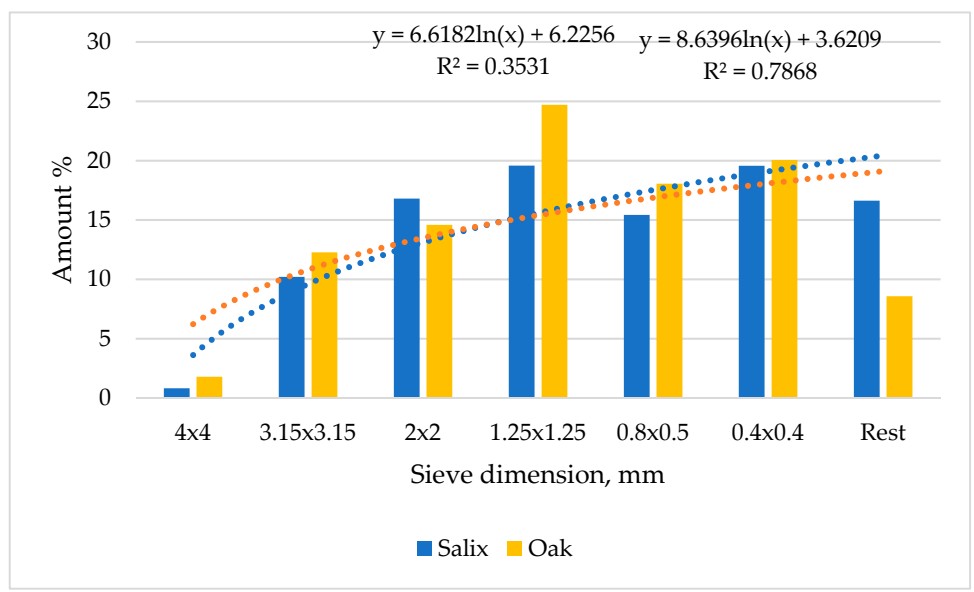

**Figure 1.** Granulometry of the crushed material for the two types of biomasses.

### 3.4. Dimensions of Briquettes and Pellets

Based on the values of the 38 samples that were analyzed, average diameter values of 40.86 mm were found in the case of briquettes and 6.35 mm in the case of pellets. The percentage coefficient of expansion after evacuation from the press was 2.1% in the case of briquettes and 5.8% in the case of pellets. The different expansions or enlargements of the pellets were explained by their increased density compared to briquettes. The length of briquettes was 22.78 mm in the case of energetic willow and 52.12 mm in the case of oak biomass, meaning that the oak biomass is more compactable.

In the case of oak, the average diameter for the 38 samples was 41.34 mm for briquettes and 6.26 mm for the pellets. The coefficient of expansion in the case of oak sawdust was 3.3% for briquettes and 4.3% for pellets. The small differences between the expansions for oak and energetic willow were due to the fact that both briquettes and pellets of energetic willow and oak were made on the same briquetting and pelletizing installations. The length of pellets was 12.0 mm in the case of energetic willow and 22.0 mm in the case of oak biomass. Related to the higher length of oak pellets, the same explanation as in the case of briquettes was identified.

### 3.5. Unit Density of Briquettes and Pellets

The unit density of the briquettes differed between those obtained from energetic willow (766.7 kg/m$^3$) and oak (877.8 kg/m$^3$); the oak briquettes were 14.5% denser than the willow ones. These differences were due to the density differences between the two wood species, the oak having a density of 675 kg/m$^3$ and the energetic willow having a density of 400 kg/m$^3$ [42]. A unit density increase of 30% was obtained in the case of oak and 91.6% in the case of energetic willow. Therefore, the densification–compaction coefficient in briquettes was 1.3 in the case of oak and 1.91 in the case of energetic willow compared to the density of the woody species and about double the density of the crushed material.

At the same pressing pressure, the crushed material with lower density will compress somewhat more, but it will not compensate for the very large difference in density between the two species. However, the density of the briquettes was very low compared to other briquettes [50], all due to the briquetting machine with a hydraulic drive with a pressure of 20 atm, with the help of which some densities much lower were always obtained related to mechanical ones with helical screw or pressure hammer. It could also be seen that the range of limiting variation of the values in the case of the energetic willow is almost double that of the oak, thus demonstrating a greater inhomogeneity.

The densities of the pellets obtained from the biomass of the energetic willow (1101 kg/m$^3$) and the oak (1296.3 kg/m$^3$) were different, with 17.7% higher in the case of the oak compared to that of the energetic willow. The explanations regarding this difference, as well as the range of variation of the different values in the case of the two types of biomasses, were similar to those in the case of briquettes, respectively; it is due to the different densities of the two wood species.

### 3.6. Pellet Torrefaction

The main parameter of pellet torrefaction was the mass loss, which increased a little with the treatment period of 1, 2, and 3 h but increased a lot with the increase in temperature from 180 to 220 °C. Therefore, maximum mass loss values were obtained for temperatures of 220 °C, between (16.78%–23.47) % for energetic willow and between (20.77–27.47) % for oak (Figure 2).

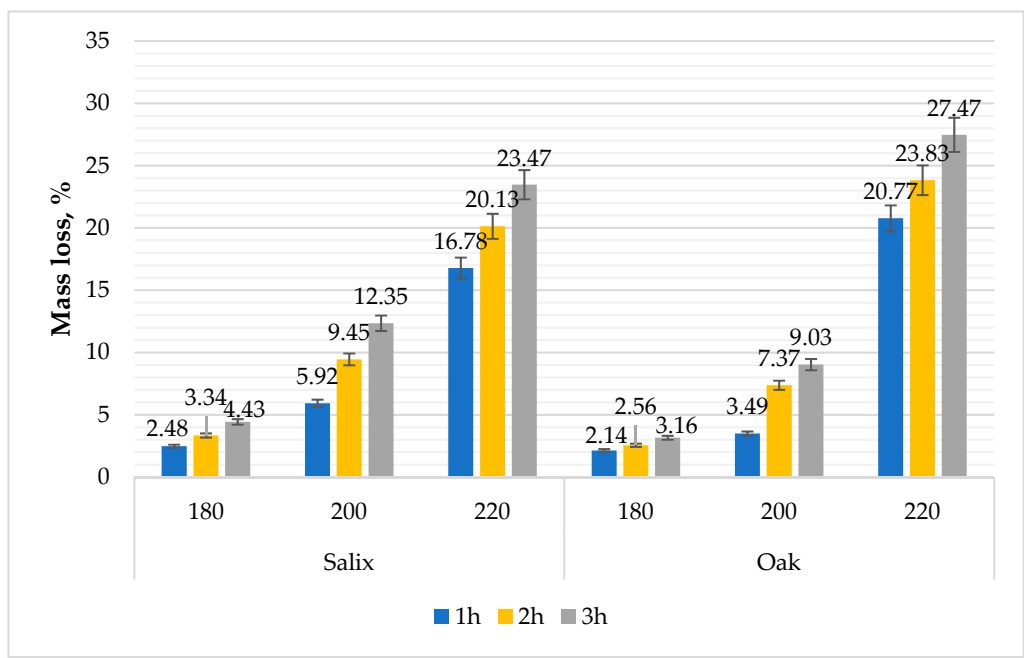

**Figure 2.** Loss of masses from torrefaction of energetic willow and oak; temperature and time are variable.

It was observed that oak was torrefied better than energetic willow, the losses being somewhat higher due to the chemical composition of the two analyzed species.

### 3.7. Calorific Value and Energetic Density

Figure 3 shows the low and high calorific values for the two analyzed species, depending on the torrefaction treatment used, respectively, depending on the three torrefaction temperatures, 180 °C, 200 °C, and 220 °C.

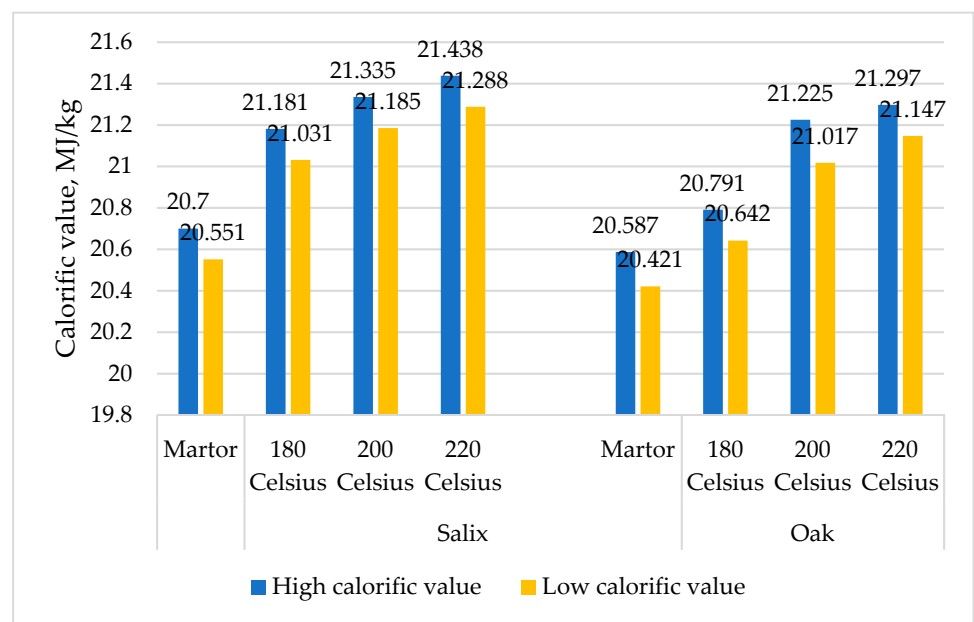

**Figure 3.** The calorific value of energetic willow and oak pellets depends on the applied thermal treatments; temperature and species are variable.

It is observed that the willow biomass had a higher calorific value than the oak biomass, regardless of the treatment applied; the increase in the calorific value of the energetic willow compared to the calorific value of the oak for the maximum treatment applied was 141 kJ/kg or a percentage of only 0.6%. In total, following the torrefaction process, the calorific value of energetic willow biomass increased by 3.5%, and that of oak increased by 3.4%.

Energetic density, definite with Equation (13), has values of $15.8 \times 10^3$ and $18.0 \times 10^3$ MJ/m$^3$ in the case of native Salix and Quercus briquettes and values of $22.7 \times 10^3$ MJ/m$^3$ and $26.67 \times 10^3$ MJ/m$^3$ in the case of native Salix and Quercus pellets. When pellets are torrefied, the energetic density slightly decreases because of higher mass losses related to the value of the calorific increase.

*3.8. The Pellets Color after the Torrefaction Process*

The torrefaction time had very little influence on the color of the pellets; the most obvious color change was observed when the torrefaction temperature was changed (Table 1).

**Table 1.** The color of the pellets in the CIELab coloristic space.

| Pellet Specie | Treatment | CIELab | | |
|---|---|---|---|---|
| | | **L\*** | **a\*** | **b\*** |
| *Salix viminalis* | Control | 36.5 | −15.3 | 5.6 |
| | 180/3 | 28.7 | −20.1 | 6.5 |
| | 200/3 | 22.8 | −21.5 | 7.1 |
| | 220/3 | 19.4 | −22.7 | 8.4 |
| *Quercus robur* | Control | 38.1 | −21.8 | 9.1 |
| | 180/3 | 36.2 | −24.5 | 9.7 |
| | 200/3 | 32.5 | −36.9 | 10.1 |
| | 220/3 | 28.5 | −47.5 | 10.9 |

It can be seen that the values of the parameter L\* have increased, which means that the pellets change their color from a light gray to black; the values of the parameter a\* are

negative, which means that the color remains in the dominant green range, and the values of the parameter b* increase slightly during torrefaction, remaining in the domain of the dominant yellow color.

### 3.9. Compressive Strength of Briquettes

As can be seen in Figure 4, the compressive strength of the two types of briquettes was very different from one to another, with willow briquettes having a compressive strength of 1.02 N/mm$^2$ and oak briquettes a compressive strength of only 0.33 N/mm$^2$. This difference highlights the fact that oak briquettes will break much faster during transport and storage in multi-stored bags.

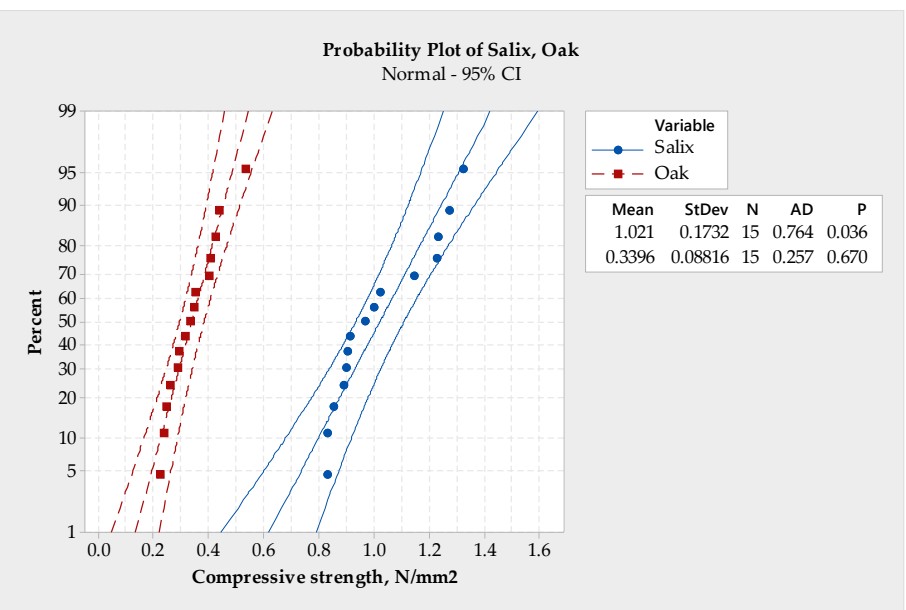

**Figure 4.** The compressive strength of briquettes: StDev-standard deviation; N-number of samples; AD-Anderson-Darling coefficient; *p*-statistic coefficient.

Statistical significance of the 15 values from Figure 4, represented by *p*-value, falls within the limiting value of 0.05 only for the energetic willow with a value of 0.036, the value of 0.67 for the oak exceeding the imposed limit. This proves that the energetic willow is more homogeneous from the point of view of compressibility. The other Anderson–Darling coefficient shows the same statistical trend.

### 3.10. The Splitting Strength of Briquettes

The splitting resistance of briquettes (similar to the splitting resistance of wood) was divided into two parts, namely, one perpendicular to the length of the briquette and the other parallel to the length of the briquette. The splitting strength perpendicular to the length of the briquette had small values, about 0.08 N/mm$^2$ in the case of oak and 0.05 N/mm$^2$ in the case of energetic willow. The standard deviation of these values was 0.009 N/mm$^2$ for oak briquettes and 0.006 N/mm$^2$ for energetic willow (Figure 5). A large variety of values is also observed.

Regarding the splitting of the briquettes, parallel to the length of the specimen (Figure 6), small average values were obtained, namely 1.0 N/mm$^2$ in the case of oak briquettes and about 0.85 N/mm$^2$ in the case of energetic willow.

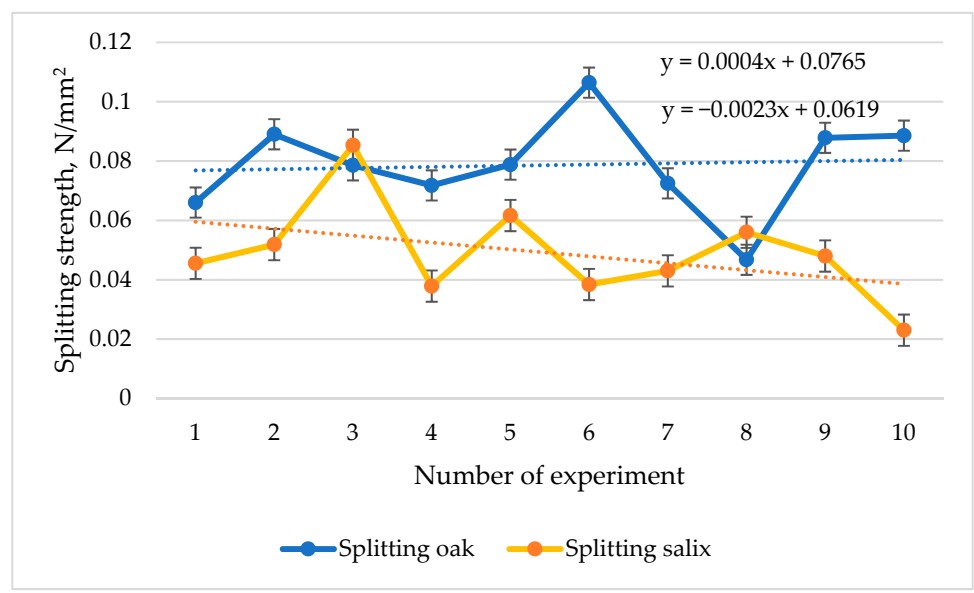

**Figure 5.** Resistance to splitting perpendicular to the length of the briquette.

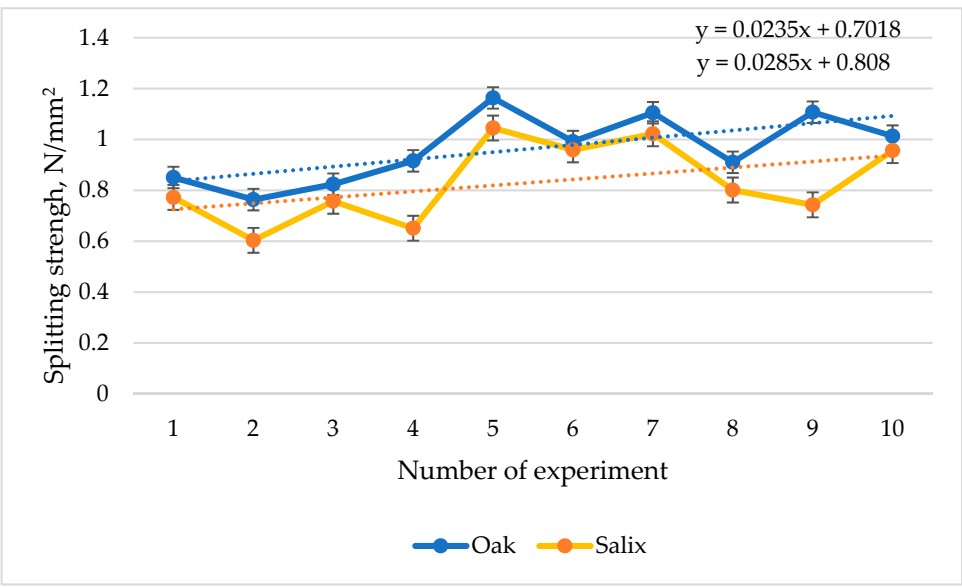

**Figure 6.** Resistance to splitting parallel to the length of the specimen.

Overall (parallel and perpendicular to the length), the splitting resistance of the energetic willow was slightly lower than that of the oak briquettes.

### 3.11. Shear Strength of Pellets

The average values of pellet shear were $0.74 \, \text{N/mm}^2$ for willow pellets and $0.86 \, \text{N/mm}^2$ for oak pellets, with standard deviations of $0.045 \, \text{N/mm}^2$ and $0.18 \, \text{N/mm}^2$, respectively. The increase in the shear resistance of the pellets in the case of oak by about 16.2% was determined by the higher density of the oak wood by 68%, from which the pellets were produced, but mainly due to the increase in the density of the pellets by about 17.7%.

Since the order of the experimental values could influence the regression equation, only the relative position of the regression equations for the two types of pellets was analyzed in Figure 7. Due to the parallelism of the two linear equations, it can be concluded that there are no statistically significant differences between the two groups of values.

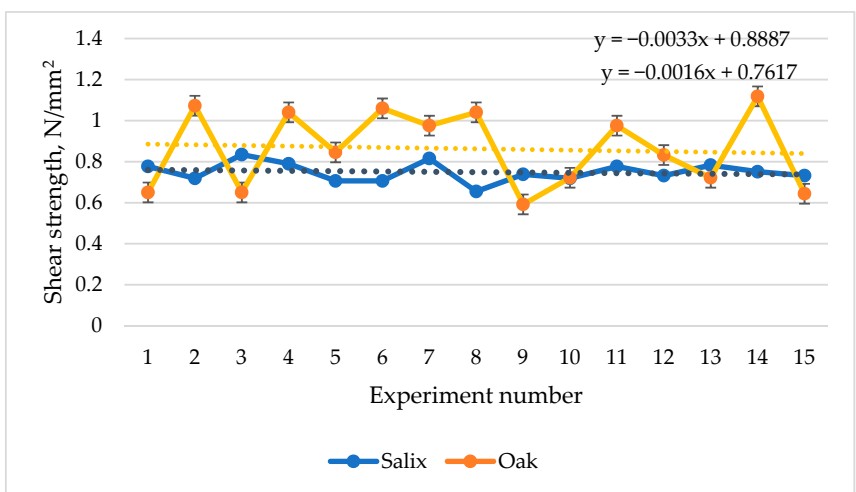

**Figure 7.** Shear resistance of energetic willow and oak pellets. Salix and oak pellets are variables.

Taking into account the average values and standard deviations of shear strength for a confidence interval of 95% or an alpha-type error of 0.05%, a value range of 0.65–0.83 N/mm$^2$ was obtained in the case of energetic willow pellets and of 0.5–1.22 N/mm$^2$ in the case of oak pellets. It is clearly observed that the range of variation of oak pellets was much wider than that of energetic willow pellets; that is, oak pellets were much more inhomogeneous. This can also be seen from the study of the standard deviation, its value in the case of willow pellets being 75% lower than in the case of oak pellets.

*3.12. The Briquette Abrasion*

The willow briquettes had an abrasion of 1.92%, and the oak briquettes had an abrasion of 4.22% (Figure 8). The 55% lower value of the abrasion of the energetic willow was due to its lower density, which caused less intensity and force when rubbed by the sieve on which the abrasion was made. Even if of lower density, the briquettes of energetic willow had a favorable result on the abrasion.

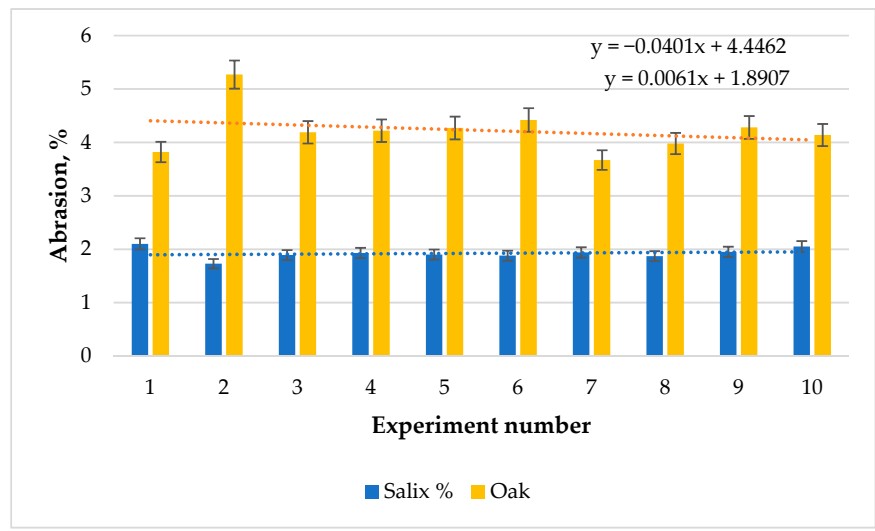

**Figure 8.** Abrasion of energetic willow and oak briquettes. Salix and oak biomass are the variables.

*3.13. Ash Content*

The black ash (Figure 9) obtained after the end of the flame had some values of over 10% for both energetic willow and oak sawdust, slightly higher in the case of oak due to substances in the form of oxides in a larger amount inside of the structure of this species.

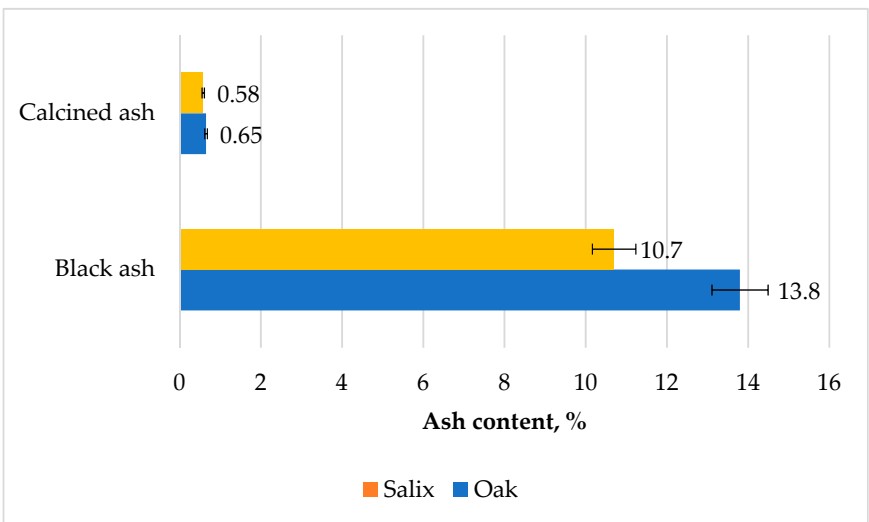

**Figure 9.** The content of calcined ash and black ash in the case of oak and energetic willow. The variable is the ash content.

The calcined ash of the energetic willow (as well as of the oak) has kept the current values of the woody species, i.e., fell under 1%. Effectively, the calcined ash content of the energetic willow was 12% lower than that of the oak, which brings an added value to this species.

*3.14. Modeling the Calorific Power Depending on the Content of Chemical Components*

This modeling starts from the premise that all lignocellulosic materials have a different calorific value of the chemical constituents, respectively, of 24.4 MJ/kg for lignin, 18.6 MJ/kg for cellulose, 16.1 MJ/kg for hemicelluloses, 34.5 MJ/kg for extractives, and 0.2 MJ/kg for ash [60–69]. The influence of the torrefaction process is also considered [35,69–71], along with other original research-specific correlations. If the different participation percentage of the chemical components for each wood species is taken into account, the following calculation relationship (Equation (18)) is obtained:

$$CV = 34.5 \times \frac{Ext}{100} + 24.4 \times \frac{Lig}{100} + 18.6 \times \frac{Cel}{100} + 16.1 \times \frac{Hem}{100} + 0.2 \times \frac{Ash}{100} \left[\frac{MJ}{kg}\right] \qquad (18)$$

where: *Ext* is the percentage of extractives from the species in %; *Lig* is the lignin percentage of the species in wt%; *Cel* is the percentage of cellulose of the species in % wt; *Hem* is the percentage of hemicelluloses of the species in % wt; *Ash* is the percentage of ash content of the species in % wt.

It is also taken into account that the degradation of hemicelluloses starts at over 180 °C, of cellulose at over 240 °C, and lignin at over 280 °C [46,50]. Therefore, during the heat treatment process by torrefaction, the hemicelluloses are degraded first, and finally, the lignin ash content remains unchanged.

Following this methodology, a calorific value was obtained for native pellets of 20.46 MJ/kg for willow and 20.33 MJ/kg for oak, with a small difference compared to the research values (Table 2). Next, taking into account the loss of mass during torrefaction over the entire thermal treatment area of 23.4% for willow and 27.4% for oak. Taking into account these mass losses, the percentages of cellulose and hemicellulose were changed. Specifically for the energetic willow, 23.4 % was subdivided into 17.09% hemicellulose, with the difference of 6.31% coming from the cellulose. Therefore, there will be a new division of the components from 100%, namely "%ref" in Table 2. The 23.4% will be divided between the other components depending on the initial percentages of the chemical compounds.

**Table 2.** Modeling the calorific power during torrefaction.

| Constituents | | Extractive 34.5 MJ/kg | Lignin 24.4 | Cellulose 18.6 | Hemicelluloses 16.1 | Ash 0.2 | Total - | MJ/kg, Equation (18) |
|---|---|---|---|---|---|---|---|---|
| Salix-native | % | 6.07 | 27.31 | 49.11 | 17.09 | 0.58 | 100 | - |
| | KJ/kg | 209.41 | 666.364 | 913.446 | 257.149 | 0.118 | 2046.48 | 20.46 |
| Salix-torrefied | % | 6.07 | 27.31 | 42.80 | 0 | 0.58 | 77.6 | - |
| | %ref. | 7.47 | 33.74 | 54.09 | 3.99 | 0.71 | 100 | - |
| | kJ/kg | 257.025 | 823.256 | 1006.074 | 64.239 | 0.142 | 21150.736 | 21.15 |
| Oak-native | % | 8.20 | 20.05 | 46.08 | 25.12 | 0.65 | 100 | - |
| | kJ/kg | 282.9 | 489.22 | 857.088 | 404.43 | 0.13 | 2033.76 | 20.33 |
| Oak-torrefied | % | 8.20 | 20.05 | 43.8 | 0 | 0.65 | (−27.4) | - |
| | %ref. | 10.44 | 25.54 | 56.42 | 6.87 | 0.82 | 100 | - |
| | kJ/kg | 360.18 | 623.176 | 1047.412 | 110.607 | 0.164 | 2141.539 | 21.41 |

These new percentages of the chemical components will provide a new calorific value of the torrefied pellets of 21.15 MJ/kg for energetic willow and 21.41 for oak. These values were appropriated to the tested value, meaning the modeling was correctly predicted.

## 4. Discussion

In order to be able to have a discussion on the properties of the analyzed briquettes and pellets, the average values from the paper were centralized together with those of other authors and standards [34,50,56] in Table 3. In addition to these data, other data were taken and analyzed from the research studies of other authors [7,27,35,41–43,57,70–72].

**Table 3.** Synthesis of properties.

| No. | Property | Own Values of Research Willow | Oak | Other Value | References |
|---|---|---|---|---|---|
| 1. | Unit density of briquettes, kg/m$^3$ | 766 | 877 | 620–720 | CRI-R0415 |
| 2. | Unit density of pellets, kg/m$^3$ | 1101 | 1296 | Min 1000 | ONORM M7135 |
| 3. | Mass loss of torrefaction, % | 23 | 27 | 5–22 | Tumuluru et al. [34] |
| 4. | Lightness (L*) of native pellets | 36.5 | 38.1 | 62 (beech) | Mitani and Barboutis [56] |
| 5. | Lightness (L*) of torrefied pellets | 19.4 | 28.5 | 45 (beech) | Mitani and Barboutis [56] |
| 6. | Calorific value for native biomass, MJ/kg | 20.7 | 20.5 | 17.5–19.5 | DIN 51731 |
| 7. | Calorific value after torrefaction, MJ/kg | 21.4 | 21.2 | 22.9 | Bi et al. [67] |
| 8. | Compressive strength, N/mm$^2$ | 1.02 | 0.33 | 0.53 | Brozek et al. [68,69] |
| 9. | Abrasion of briquettes, % | 4.22 | 1.92 | (1.5–3)% | SS 18 17 20 |
| 10. | Ash content, % | 0.58 | 0.62 | Max 6% | ONORM M7135 |

The unit density of energetic willow briquettes (766 kg/m$^3$) was 12.5% lower than that of oak; this difference is due to the 68.7% increased density of oak wood compared to that of energetic willow biomass. Referring to the limiting values of 620–720 kg/m$^3$ of the Italian Standard CRI-R0415, it can be seen that the experimental values fall within these limits. In addition, Brozek et al. [68] found a density of poplar briquettes of 776 kg/m$^3$ and 692 kg/m$^3$ for briquettes from birch biomass. The unit density of pellets, as 1001 kg/m$^3$ for Salix and 1269 kg/m$^3$ for oak, falls within the limiting provisions of the Austrian Standard ONORM M7135 (minimal 1000 kg/m$^3$). Similar values were found by other authors [27,28].

The experimental values of 23% and 27% mass loss by torrefaction are also confirmed by Tumuluru et al. [34], who found values of 5%–22% (for temperatures of 230–270 °C) for willow and 4%–14% for rice straw and wheat. The torrefaction treatment destroys OH groups in the wood, forming a non-polar unsaturated chemical structure, which will protect the pellets against biological degradation, similar to charcoal [56].

The values of the luminance (L*) expressed in the CIELab space of the native pellets obtained from the biomass of the energetic willow and the oak were different, with a difference of 9.1 color units. From this point of view, the energetic willow had a color closer to white than the oak. After torrefaction, the luminance of the two categories of pellets is very close, the difference being only 1.6 color units, in a shade of gray much closer to black than native pellets. Mitani and Barboutis [56] found that the beech species has a decrease in luminance during the torrefaction treatment from 60 to 45 color units, the difference depending on the treatment temperature but also on the structural direction of the wood (longitudinal, radial, and tangential) which is taken into consideration.

The calorific value of native energetic willow (20.7 MJ/kg) and native oak biomass (20.5 MJ/kg) fell within the provisions of all European reference standards, respectively, higher than 16.9 MJ/kg (SS 18 17 20), higher than 18 MJ/kg (ONORM M 7135), between 17.5–19.5 (DIN 51731:2000) and higher than 16.2 MJ/kg (CTI-R0615). Moreover, the calorific value of torrefied pellets, higher than that of native pellets, falls within the above limiting values of European standards. The values obtained in this study are slightly higher than those obtained by other researchers [69] for the clone of the energetic willow *Salix shwerinii* of 20.02 MJ/kg. The same authors found other calorific values of 20.2 $N/mm^2$ for *Alnus glutinosa* and 19.4 MJ/kg for *Betula pendula*. Balaban and Uçar [60] found an average high calorific value of 20 MJ/kg. Krajnic [42] has stated typical values of calorific value lower than 19.2 for coniferous and 19 MJ/kg for deciduous species. Referring to the calorific value for torrefied pellets, Bi et al. [67] found a maximum value of 22.9 MJ/kg, higher than the one from our own research due to the no-oxygen content during torrefaction. In addition, Hu et al. [69] found a difference between the HCV value before and after torrefaction of 2.12 MJ/kg, a value higher than the one found in the research of 0.8 MJ/kg, explained by the duration and temperature of the heat treatment performed.

Regarding the compressive strength, Brozek et al. [69] have established the compressive strengths of briquettes appropriate to those found in the research (1.02 $N/mm^2$ for energetic willow and 0.33 $N/mm^2$ for oak). For example, taking into account the diameter of the briquettes of 60 mm, for briquettes obtained from poplar biomass, a value of 1.35 $N/mm^2$ was found, and for biomass from the bark of the same species, a value of 0.53 $N/mm^2$ was also obtained.

The abrasion determines the amount of dust resulting from the transport and handling of briquettes, established by the Swedish Standard SS 18 17 20 in the form of "fines" smaller than 3 mm at a range of (1.5%–3%). The values of 4.22% for the energetic willow exceed the maximum value of 3% due to the low density of the briquettes. The abrasion value of 1.92% for the oak biomass briquettes falls within the limits of the standard, especially due to the density of 877 $kg/m^3$, 14.4% higher than that of the energetic willow.

The limits of the calcined ash content are different from one standard to another, being (0.7%–1.5%) for woody biomass (SS 18 17 20, DIN 51731 and CTI-R0415) and a maximum of 6% for agricultural biomass (ONORM M7135). Agricultural biomass has a higher mineral content than wood, which is why the ash content is also higher [67–69]). The experimental values of 0.58% for *Salix viminalis* and 0.62% for *Quercus robur* fall within the limits of European standards. Other researchers found values higher than the standardized limits for *Populus alba* (3.61%) and its bark (1.72%) [67], and others [69] found a value of 0.44% for the *Salix shwerinii* clone, 2.17% for *Populus tremula*, 0.78% for *Alnus glutinosa*, and 0.75% for *Betula pendula*.

## 5. Conclusions

- Energetic willow has a calorific value of 20.7 MJ/kg and an energy density of $22.7 \times 10^3$ $MJ/m^3$, higher than those of oak of 20.58 MJ/kg and $26.6 \times 10^3$ $MJ/m^3$, respectively. So, the energetic willow has very good calorific behavior, its properties being better than those of oak and other wood species used in the energetic field. Moreover, the research demonstrated why willow is considered one of the deciduous species with the highest calorific value.

- The torrefaction treatment at maximum regime led to a better calorific value of 21.43 MJ/kg in the case of energetic willow, compared to only 21.29 MJ/kg in the case of oak.
- The calcined ash content was lower in the case of energetic willow, with a value of 0.59%, compared to 0.65% in the case of oak.
- Ecologically, energetic willow has the same positive effects of sequestering carbon dioxide from the air and releasing oxygen as any other fast-growing woody species used in combustion.
- Future research will be focused on increasing the carbon content of the energetic willow biomass during the torrefaction process.

**Author Contributions:** Conceptualization, A.L. and V.D.; methodology, C.S. (Cosmin Spirchez); software, C.S. (Cezar Scriba); validation, A.L., C.S. (Cosmin Spirchez) and C.S. (Cezar Scriba); formal analysis, C.S. (Cezar Scriba); investigation, C.S. (Cosmin Spirchez); resources, C.S. (Cezar Scriba); data curation, C.S. (Cezar Scriba); writing—original draft preparation, A.L.; writing—review and editing, A.L.; visualization, C.S. (Cezar Scriba); supervision, A.L.; project administration, A.L.; funding acquisition, C.S. (Cosmin Spirchez) All authors have read and agreed to the published version of the manuscript.

**Funding:** This research received no external funding.

**Data Availability Statement:** Not applicable.

**Conflicts of Interest:** The authors declare no conflict of interest.

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
