# Peer review of "Some Properties of Briquettes and Pellets Obtained from the Biomass of Energetic Willow (Salix viminalis L.) in Comparison with Those from Oak (Quercus robur)"

_forests, doi:10.3390/f14061134_

Round 1

Reviewer 1 Report

I have this article and the result could be beneficial to briquette production. however, the biggest challenge is the quality of overall presentation which is very poor. Thus, I make the following suggestion.

1. The authors require an English editor to technical edit and re-organize their work before further resubmission.

2. the introduction lacks co-ordination, various section was misplaced and reading it is difficult. Additionally, the introduction requires a lot of current literature.  captioning the objective differently should be removed and submerge it in the introduction at the last paragraph. Also, the authors should show vigorously the contribution of their work to advance the existing knowledge.

3. Equations should be presented with equation editor and properly formatted.

4. there is need to re-write the entire methodology in a technical and scientific manner for clarity and for it to flow to the reader

5. I suggest the authors merge result and discussion together and use existing literatures to extensively discuss their work

6. section 3.1 contains a lot of loose words and unnecessary deductions. this should be curtailed, and the actual result obtained discussed in comparative way with other similar plants in terms of the value of their carbon mitigation potentials.

7. variables in regression equations should be defined

8. I suggest a nomenclature section to be added.

9. graphs should be replotted with different marker shapes to differentiate them in black and white printing

10. I don't think figure 2 and 3 is relevant. Why plot individual number or observation. it is not explanatory.

11. itemize your conclusions and give the future research perspectives.

I have read this article and the result could be beneficial to briquette production. however, the biggest challenge is the quality of overall presentation which is very poor. Thus, I make the following suggestion.

The authors require an English editor to technical edit and re-organize their work before further resubmission.

Author Response

Reviewer 1.

We would like to thank the reviewer for the submitted activity.

  1. The authors require an English editor to technical edit and re-organize their work before further resubmission.

Authors response: Our work was once again checked from the point of view of the grammatical structures of the English language by an English language teacher of our university specialized in this professional field, and repaired accordingly. Please accept this economic solution for authors. Also, the revised paper respects much more the formation of Forests journal.     

  1. the introduction lacks co-ordination, various section was misplaced and reading it is difficult. Additionally, the introduction requires a lot of current literature.  captioning the objective differently should be removed and submerge it in the introduction at the last paragraph. Also, the authors should show vigorously the contribution of their work to advance the existing knowledge.

Authors response: The authors made a lot of changes in the paper and we hope these modifications are accordingly to reviewer requirements.  

  1. Equations should be presented with equation editor and properly formatted.

Authors response:  All equations were written by Microsoft equation (the first 7 items).

  1. there is need to re-write the entire methodology in a technical and scientific manner for clarity and for it to flow to the reader

Authors response: When we rewrote and corrected, we took these considerations into account, i.e., we took into account that the reader could easily read the research, that is, the writing should be clear, concise and to the point.  

  1. I suggest the authors merge result and discussion together and use existing literatures to extensively discuss their work

Authors response: We fully agree with this request of the reviewer, because in this way the reader would find all the information in the same place and would understand the information presented more quickly. Unfortunately, the editors of the journal oblige this format of the work, and we authors must respect this format of the journal. See   https://www.mdpi.com/journal/forests/instructions.  Otherwise, if we did not respect the formatting template of the magazine, the editors would ask us to reformat, or the paper would not have any chance to move forward. That's why we ask you to accept this formatting of the work.

  1. section 3.1 contains a lot of loose words and unnecessary deductions. this should be curtailed, and the actual result obtained discussed in comparative way with other similar plants in terms of the value of their carbon mitigation potentials.

Authors response: The problem of carbon sequestration from nature by trees is a general and almost similar one. In the work, an attempt was made to find the elements that individualize the potential of the energetic willow.

  1. variables in regression equations should be defined

Authors response: For each graph with regression equation, their variables were entered.

  1. I suggest a nomenclature section to be added.

Authors response: The introduction of all the specific terms in the paper as well as abbreviations would add at least two pages to the paper, which is already very long with 23 pages. Also, the fact that each term is defined within the equations, in tables or methodologies, would make them redundant. Last but not least, the journal template does not allow such a nomenclature table. Therefore, please allow us not to make this addition.

  1. graphs should be replotted with different marker shapes to differentiate them in black and white printing.

Authors response: All the graphics have been changed from this point of view, respectively the red color has changed to orange. 

  1. I don't think figure 2 and 3 is relevant. Why plot individual number or observation. it is not explanatory.

Authors response: The two figures are important for the paper, highlighting the arithmetic mean, the lower and upper limit of the group of values. In this way, both the general trend of the values and the dispersion of the values are identified. Also, for any reader, the message is easier to understand on a graph than from a written paragraph. That is why we ask for your permission for the two figures to remain in the work.    

  1. itemize your conclusions and give the future research perspectives.

Authors response: Conclusions were individualized by bulleted, in order to be more visible. Also, a new conclusion was added related to the future researches: “Future research will be focused on increasing the carbon content of the energetic willow biomass during the torrefaction process“.

Authors,

Reviewer 2 Report

The authors presented research related to the evaluation of the importance of plantation biomass for energy purposes. The importance of the need to suspend the use of plantation crops was indicated. Changes in the energy value of plantation biomass from energy willow were presented. The authors interestingly presented studies for comparable samples of energy willow briquettes and oak.

The relevance of obtaining solid fuels with suitably modified composition in order to reduce the negative impact of the use of fossil fuels and CO2 emissions was indicated.

Notes

The introduction does not present all important issues. What are the economic effects and what is the payback period for fast-growing trees? What is the reference to other fast-growing plantation species? No reference to literature.

In the discussion of the categories of energy crops (line 45 et seq.) there is no direct reference to the literature.

The indication that the energy obtained is renewable energy, for young plantations refers to items of several clustered items :8-12_( Line 66). Is it possible to make a more precise reference for the results presented in the given publications. This will be a more precise form of information transfer.

The presentation of the process of cultivation of energy truss referred to numerous publications - 13-32! (line 70). It will be better if the authors indicate the influence of individual publications in the description of the cultivation of the presented species.

The division of the literature in line 91 seems inappropriate 40-44 and 45-47 ? would it not be appropriate to present separate results from the range 40-44 and separately 45-47?

The methodological description is elaborate and based on mathematical assumptions.

In the case of the description of the methodology of obtaining the experimental material: the divisions and information (designations) on the separated groups with a variable proportion of dimensional fractions directed to further studies were not clearly presented . The types of sieves are given, how they are related to the designation of further trials - there is no designation for the assigned sample of 7 types of particle size directed to the production of briquettes.

For the separated groups of energy willow and oak biomass, it is indicated to obtain chips and briquette production for further study. What is the rationale for the study of caloric improvement by treatment at 180, 200 and 220 C for variable times of 1, 2 and 3 hours?  This was based solely on the literature reference (item 35). What is the reference to other test methods related to calorimeter tests?

Calorimeter tests and briquette hardness tests were described accurately.

10 tests were conducted for willow and oak biomass. What are the statistics of the results for the tested batches of material?

Figure 2 and 3 present 37 observations - what is the effect of the order of observations on the results obtained - why do the authors combine ndividual Value points on the graph? Since these are briquette densities?

Figures 7-10 show the number of experiments. What is the significance of the order of the excrement, no explanation of the significance of the next experiment if it differed in biomass characteristics. If there is no change in parameters is just a data set from one test batch.

Kindly clarify and complete the information in the description.

In Table 2, the authors present the modeling of calorific power during torrefaction of chemical components of biomass - what is the basis (sources) of literature or research underlying the modeling ?

Table 3 presents a synthesis of properties. What is the literature reference for the standards presented. Other references find a place in References.

The discussion and conclusions are synthesized and presented correctly.

The graphs need to be corrected to better present the results.

Statistical significance assessment of the presented research results was missing

Literature needs corrections e.g.

Line 680 where Is : Mleczeke, P, - should be Mleczek, P.

And other similar ...

The article is interesting and important from the point of view of assessing the need for renewable energy. I look forward to corrections and clarifications.

Author Response

Reviewer 2

We would like to thank the reviewer for the submitted activity.

  1. The introduction does not present all important issues. What are the economic effects and what is the payback period for fast-growing trees? What is the reference to other fast-growing plantation species? No reference to literature.

Authors response: We add the next paragraph: “A plantation with ordinary woody species of trees needs on average 90-160 years to reach maturity, and a plantation with fast-growing species needs only 25-35 years. That's why, besides the fact that plantations with fast-growing species recover the investment faster, they can obtain a 72-78% better economic efficiency”. We have 12 references about short-rotation coppice.

  1. In the discussion of the categories of energy crops (line 45 et seq.) there is no direct reference to the literature.

Authors response: We put a reference in the specified place.

  1. The indication that the energy obtained is renewable energy, for young plantations refers to items of several clustered items: 8-12 (Line 66). Is it possible to make a more precise reference for the results presented in the given publications? This will be a more precise form of information transfer.

Authors response: The compact citations were divided into several parts, taking into account that the information corresponds to the citation. In this way, the information will be more appropriate to the citations.

  1. The presentation of the process of cultivation of energy truss referred to numerous publications - 13-32! (line 70). It will be better if the authors indicate the influence of individual publications in the description of the cultivation of the presented species.

Authors response: We erase and complete this paragraph with different approach of referenced authors. There is a new paragraph: “A lot of authors treated climate change [13], financier analysis [14], energetic impact [15], biofuel potential [16], yield [17-20], establishing of surface cover [21-22], biomass production [23], trait and genome [24-25], potential [26], wood quality [27], briquetting [28], genetic structure and diversity [29-30], structure [31], and prognosis [32], all of these being obtained by energetic willow plantation.”

  1. The division of the literature in line 91 seems inappropriate 40-44 and 45-47 ? would it not be appropriate to present separate results from the range 40-44 and separately 45-47?

 Authors response: We change that paragraph and become: “They keep the advantages of lignocellulosic biomass, including those related to renewability [40], environmental friendliness [41] and neutrality carbon dioxide emission [42]. Also, briquettes obtained from energetic crops (miscanthus, energy willow, sorghum, etc.) [43], bring an important contribution to the natural environment by eliminating oxygen [44] and sequestering carbon dioxide in each vegetative year [6; 45]. Other economic and ecological effects of biomass briquetting are presented by other researchers [46-47].”

  1. In the case of the description of the methodology of obtaining the experimental material: the divisions and information (designations) on the separated groups with a variable proportion of dimensional fractions directed to further studies were not clearly presented. The types of sieves are given, how they are related to the designation of further trials - there is no designation for the assigned sample of 7 types of particle size directed to the production of briquettes.

Authors response: We add a new paragraph, for more explanation of the method. “Both types of small materials were sorted with a 5x5 mm sieve, in order to have appropriate and homogeneous sizes. The main purpose of this determination was to determine the different fractions of the small material, because an increased percentage of the fraction with large sizes will determine a low density and a high breaking strength of the briquettes and pellets, and an increased percentage of the fraction with small sizes will lead to obtaining some products with high densities and reduced resistance.”

  1. For the separated groups of energy willow and oak biomass, it is indicated to obtain chips and briquette production for further study. What is the rationale for the study of caloric improvement by treatment at 180, 200 and 220 C for variable times of 1, 2 and 3 hours?  This was based solely on the literature reference (item 35). What is the reference to other test methods related to calorimeter tests?

Authors response: In order to be more understanding the aim of method, a new paragraph was added. “During torrefaction, part of the wood hemicelluloses is damaged, thereby increasing the calorific value but especially the energetic density through the loss of mass. Additionally, the torrefied pellets become more stable (moisture absorption is reduced) and are sterilized (degradation is more difficult). The higher temperature and duration, the advantages of torrefaction will be.” Other references were added [41-42].   

  1. 10 tests were conducted for willow and oak biomass. What are the statistics of the results for the tested batches of material?

Authors response: We add a new paragraph. “Standard deviation of these values was 0.009 N/mm2 for oak briquettes and 0.006 N/mm2 for energetic willow. A variety of values is also observed.”

  1. Figure 2 and 3 present 37 observations - what is the effect of the order of observations on the results obtained - why do the authors combine individual Value points on the graph? Since these are briquette densities?

Authors response: We add a new paragraph. “In figures 2 and 3, the order of presentation of the values has no influence, because the 3 parameters are determined (average, upper and lower limit), for a confidence interval of 95%. This is why this type of chart was used.”

  1. Figures 7-10 show the number of experiments. What is the significance of the order of the excrement, no explanation of the significance of the next experiment if it differed in biomass characteristics? If there is no change in parameters is just a data set from one test batch. Kindly clarify and complete the information in the description.

Authors response: We add a new paragraph: “Since the order of the experimental values could influence the regression equation, only the relative position of the regression equations for the two types of pellets was analyzed in Figure 9. Due to the parallelism of the two linear equations, it can be concluded that there are no statistically significant differences between the two groups of values.” Also, referring to significance, we add a new paragraph: “Statistical significance of the 15 values from Figure 6, represented by p-value falls within the limiting value of 0.05 only for the energetic willow with a value of 0.036, the value of 0.67 for the oak exceeding the imposed statistical limit. This proves that the energetic willow is more homogeneous from the point of view of compressibility. The other Anderson-Darling coefficient show the same statistical trend.”     

  1. In Table 2, the authors present the modelling of calorific power during torrefaction of chemical components of biomass - what is the basis (sources) of literature or research underlying the modelling?

Authors response: Additions were completed to the work. “…for ash [60,62,64,67]. The influence of torrefaction process is also taken into consideration [35,69-71], along with other original research-specific correlations.” Also, even if initially we did not want this, we added our own work in the analyzed field, reference [71].

  1. Table 3 presents a synthesis of properties. What is the literature reference for the standards presented? Other references find a place in References.

Authors response: We add a new sentence. “Besides these, other data were taken and analyzed from the researches of other authors [7, 27, 35, 41-43,57, 70-72].” Two new references were added, one of which [72] contains a synthesis of the standards used in the work.  

  1. The graphs need to be corrected to better present the results.

Authors response. All graphs were completed with standard deviation, and other statistical elements.

  1. Statistical significance assessment of the presented research results was missing

Authors response: For resolve this requirement, we added a new paragraph. “Statistical significance of the 15 values from Figure 6, represented by p-value falls within the limiting value of 0.05 only for the energetic willow with a value of 0.036, the value of 0.67 for the oak exceeding the imposed statistical limit. This proves that the energetic willow is more homogeneous from the point of view of compressibility. The other Anderson-Darling coefficient show the same statistical trend.”

Literature needs corrections e.g.

Line 680 where Is: Mleczeke, P, - should be Mleczek, P. And other similar ...

Authors response: We corrected these words.

Authors,

Round 2

Reviewer 2 Report

Thank you very much to the Authors for their considerable work in improving the article. 

All my comments have been answered. Significant additions and corrections have been made to the article

However, I am still concerned about the presentation of the results in Figures 2 and 3. Connecting the points presenting the findings with a solid line raises concerns about misinterpretation of the presented results. Trying to interpolate can be a problem without evaluating outliers and assessing the trend in the variability of the study results. Hence the recommendation for corrections in the presentation of figures and their description.

The work is valuable and the research conducted important for the assessment of the energy value of wood biomass. 

Author Response

Thank you very much to the Reviewer for its considerable work in improving the article, to make it more accessible to readers and to considerably improve its content. 

However, I am still concerned about the presentation of the results in Figures 2 and 3. Connecting the points presenting the findings with a solid line raises concerns about misinterpretation of the presented results. Trying to interpolate can be a problem without evaluating outliers and assessing the trend in the variability of the study results. Hence the recommendation for corrections in the presentation of figures and their description.

Authors response: Taking into account the reviewer's recommendation, figures 2 and 3 were permanently removed from the paper. The adaptation of the content of the whole results with the absence of the two figures was also carried out. In addition, small changes to the English language have been completed. The old changes were kept, and the new ones were also highlighted in yellow.

Authors,
